# Perceptions about COVID-19 preventive measures among Ghanaian women

**Frank Kyei-Arthur**[1][*], **Martin Wiredu Agyekum**[2], **Grace Frempong Afrifa-Anane**[1], **Reuben Tete Larbi**[3], **Peter Kisaakye**[4]

**1** Department of Environment and Public Health, University of Environment and Sustainable Development, Somanya, Ghana, **2** Institute for Educational Research and Innovation Studies, University of Education Winneba, Winneba, Ghana, **3** Regional Institute for Population Studies, University of Ghana, Legon, Ghana, **4** Department of Population Studies, Makerere University, Kampala, Uganda

☯ These authors contributed equally to this work.

* fkyei-arthur@uesd.edu.gh

**Data Availability Statement:** All relevant data are within the manuscript and its Supporting Information files.

**Funding:** The author(s) received no specific funding for this work.

## Abstract

Though the advent of COVID-19 vaccines has significantly reduced severe morbidity and mortality, infection rates continue to rise. Therefore, adhering to COVID-19 preventive measures remains essential in the fight against the pandemic, particularly in Africa, where vaccination rates remain low. However, the perceived risk associated with COVID-19 and public education and awareness campaigns has waned over time. COVID-19 vaccine hesitancy is consistently high among women globally. This study, therefore, assessed the facilitators, and barriers to adherence to COVID-19 preventive measures. A qualitative descriptive study was conducted among Ghanaian women. Twenty-seven in-depth interviews were conducted with women in the Greater Accra and Ashanti regions. All interviews were audio-recorded and transcribed verbatim into English. The data were analysed using NVivo 10 software. While some participants found the use of face masks as the easiest, others found it as the most difficult. In addition, institutional and policy decisions such as access to water and the use of public transport impacted individual level adherence to preventive measures. In conclusion, the fight against COVID-19 is not over; hence public education and the provision of facilities that would enhance compliance with preventive measures should continue to be prioritised.

## Introduction

Since the emergence of COVID-19 in November 2019, it has continued to cause havoc to individuals, households, and the global economy [1–3]. As of May 27 2022, about 525.4 million people worldwide have been infected with COVID-19, and 6.2 million have died due to COVID-19 [4]. In Ghana, 161,370 people have been infected with COVID-19, and 1,445 died as May 27 2022 [4]. Respiratory droplets and direct contact are the primary means through which COVID-19 is spread [5].

Studies have recommended a combination of measures to help reduce the spread of COVID-19, including adherence to COVID-19 preventive measures (such as wearing of face

**Competing interests:** The authors have declared that no competing interests exist.

masks, hand washing, using alcohol-based hand sanitizer, and avoiding touching eyes, mouth, or nose with unclean hands) and vaccination against COVID-19 since one single measure is not enough to combat the disease [6–8].

Currently, there is an inequitable distribution of COVID-19 vaccines, especially in the middle- and low-income countries [9–11]. Also, studies have established that SARS-CoV-2, the COVID-19-causing virus, can escape immunity, which can cause infection among people or reinfection among people who have already been vaccinated against COVID-19 or previously been infected with COVID-19 [6, 12–14]. Therefore, it is crucial to promote adherence to the COVID-19 preventive measures since studies have established that they effectively reduce COVID-19 infections [15–17]. For instance, World Health Organisation and UNICEF [5] have emphasised that hand hygiene practices (such as washing hands with soap and water and rubbing hands with alcohol-based hand sanitizer) effectively reduce the transmission of COVID-19.

Globally, there have been studies on factors associated with adherence to COVID-19 preventive measures [18–23] and facilitators and barriers to the implementation of COVID-19 measures [24]. However, there is representation paucity in literature from sub-Saharan Africa. In the Ghanaian context, available studies have focused on physical distancing and risk of COVID-19 [25], hand hygiene and safety [26], and adherence to COVID-19 preventive measures [27].

The extant literature suggests that factors that motivate adherence to COVID-19 preventive measures are multifaceted. These include individual level factors such as age, education, gender, income, knowledge about COVID-19, forgetfulness, feeling uncomfortable and fear of death [22, 28–30]. In addition, environmental and structural factors such as shortage of personal protective equipment, inadequate facilities, inaccessible face mask and hand sanitizers, feeling a responsibility to protect one's family, misconceptions about COVID-19, religious beliefs have been reported [31–33].

Studies have established that women are less willing to vaccinate against COVID-19 [34–38]. Therefore, it is critical to understand factors that drive them to comply or hesitate to COVID-19 preventive measures since their adherence to these measures has implications for the spread of COVID-19. This study, therefore, seeks to explore women's lay knowledge, facilitators and barriers to their adherence to COVID-19 preventive measures.

## Theoretical framework

Theoretically, this study was guided by the socio-ecological model which provides a comprehensive understanding of health promoting behaviours by taking into account five levels of influence: intrapersonal, interpersonal, institutional, community and public policy [39]. Fig 1 shows the five levels of the socio-ecological model. The socioecological model is useful in providing context-specific interventions and thus findings from this study will help develop interventions to promote compliance with COVID-19 preventive measures among Ghanaian women. We conceptualised the five levels of influence as follows.

At the intrapersonal level, knowledge about COVID-19 and its preventive measures, beliefs about COVID-19 and perceived risk of contracting the disease could inform adherence [40, 41]. At the interpersonal level, adherence to the COVID-19 preventive measures could be influenced by living with family members who have been diagnosed of or are more vulnerable to COVID-19 infection and the desire to protect others [34, 42]. Institutional factors such as religious beliefs could also inform adherence to the preventive measures [33]. At the community level, cultural values and beliefs about COVID-19 could have an influence on adherence of the preventive measures [32, 33]. Finally, policy decisions such as mandatory wearing of

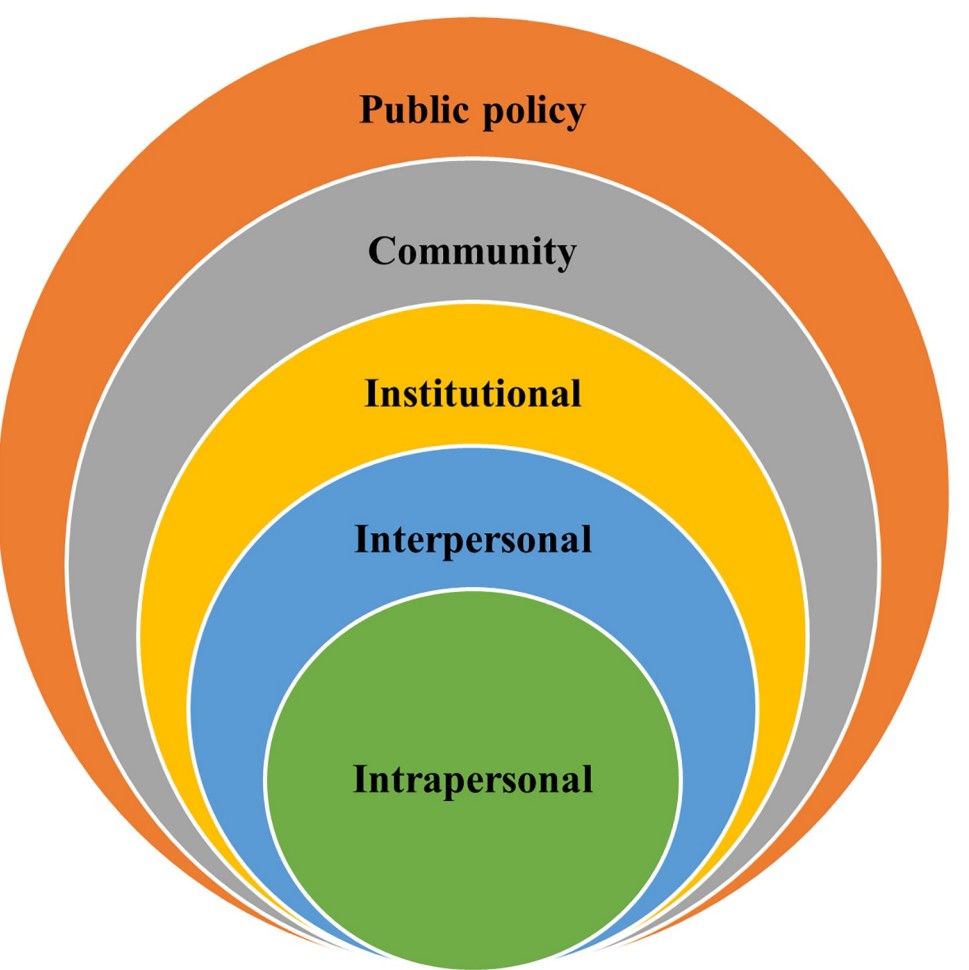

**Fig 1. The levels of the socio-ecological model applied to this study.**

face mask, washing of hands, use of sanitizers, awareness creation, and accessibility of equipment of could influence adherence to COVID-19 preventive measures [43, 44].

## Methods

### Study design and sampling procedure

This study used qualitative descriptive research design to explore women's lay knowledge, facilitators and barriers to their adherence to COVID-19 preventive measures. Qualitative descriptive research design is an interpretive approach used to summarise the experiences of persons or groups of persons extensively [45, 46]. We used a convenient sampling procedure to recruit non-pregnant women aged 18 years or older in the Greater Accra and Ashanti Regions who were willing to participate in this study. A convenient sampling procedure is a non-probability sampling, where a researcher selects participants based on their availability and willingness to participate in the study. This approach of selecting participants is appropriate for this study as the objective was to assess the attitudes and opinions of women at the selected geographical areas. Participants were selected randomly at the central business areas to decrease potential bias in the sampling process. This provided a scientifically rigorous access to participants under the constraints of low budget and lack of sampling frame for the target

population [47]. We purposively selected the Greater Accra and Ashanti regions as they are the two largest and cosmopolitan regions in Ghana [48]. In addition, infection rates were highest in these regions [34, 38, 49].

## Study setting

Ghana has sixteen administrative regions, including the Greater Accra and Ashanti regions. The Greater Accra and Ashanti regions are Ghana's most populous regions. The Greater Accra region, the capital of Ghana, has a population of approximately 5.5 million, while the Ashanti region has about 5.4 million [48]. Both regions are the most urbanised in Ghana. However, the Greater Accra region is more urbanised (91.7%) than the Ashanti region (61.6%). The Greater Accra and Ashanti regions are the epicentres of the COVID-19 pandemic in Ghana. Furthermore, the Greater Accra and Ashanti regions have more health amenities than other regions in Ghana [50].

## Data collection

The in-depth interview guide administered to women covers background characteristics, knowledge about COVID-19, and adherence to COVID-19 preventive measures (S1 File). The data collection for the study took place between October and November 2021. Twenty-seven in-depth interviews were conducted face-to-face: thirteen in the Greater Accra region and fourteen in Ashanti region. We stopped interviewing after achieving meaning saturation in both locations, and additional data did not add any new themes to the narratives. According to Guest et al. [51], twelve interviews are adequate to achieve data saturation in qualitative research. A day training workshop was organised to train Research Assistants and the first author on the interview guide. The interview guide was piloted with women in the Greater Accra and Ashanti regions, and questions that were not clear to participants were reframed for easy understanding.

The in-depth interviews were conducted in both Akan and English languages as preferred by participants, and they lasted about 30 minutes on average. Participants were informed about the nature and objectives of the study, and they gave written consent before the interviews. Participants were interviewed at their homes by the first author and Research Assistants who ensured confidentiality and privacy during the data collection. All twenty-seven in-depth interviews were audio-recorded and transcribed verbatim into English.

The Ethics Committee of the University of Environment and Sustainable Development in Somanya, Ghana, approved this study (APP/RSC/0001). All activities of the study were in accordance with ethical guidelines.

## Data analysis

The data for the study were analysed thematically using NVivo 10 software following steps recommended by Braun and Clarke [52, 53]. First, the first author read and re-read the transcripts to familiarise himself with the data and recorded ideas (codes) that emerged from the study. Second, the diverse interesting ideas (codes) across the thirty transcripts were collated. Third, similar interesting ideas (codes) across the transcripts were collated and merged into themes. Fourth, the first author examined the collated themes and merged similar themes to form one theme. Also, the first author re-read the transcripts to ensure the themes aligns with the transcripts and all interesting ideas (codes) associated with each theme are captured. Fifth, each theme was re-examined to ensure their names adequately reflect the interesting ideas (codes) under them. Sixth, the first author wrote the study's findings, a thick description of each theme. Braun and Clarke recommended steps for analysing qualitative data were followed

because it helps researchers to analyse qualitative data systematically and in a rigorous manner.

All co-authors verified the data analysis processes to ensure the trustworthiness of the findings. Also, all co-authors read the transcripts and verified the results to ensure that they reflect participants' experiences.

# Results

## Characteristics of participants

Table 1 shows the socio-demographic characteristics of participants. The average age of participants was 38 ± 13.7 years. A little over half of the participants (51.9%) had tertiary education, while less than half (48.2%) had never been married. Also, one-third of the participants (33.3%) were employed as professionals, and services and sales workers. All participants

**Table 1. Socio-demographic characteristics of participants.**

| Characteristics | Number of participants (n = 27) | Percentage |
|---|---|---|
| **Age** | | |
| Range | 20–67 | |
| Mean ± Standard deviation | 38 ± 13.7 | |
| **Education** | | |
| Primary | 3 | 11.1 |
| Senior High | 10 | 37.0 |
| Tertiary | 14 | 51.9 |
| **Marital status** | | |
| Never married | 13 | 48.2 |
| Currently married | 9 | 33.3 |
| Formerly married | 5 | 18.5 |
| **Religion** | | |
| Christian | 27 | 100.0 |
| **Occupation** | | |
| Administrator | 1 | 3.7 |
| Professional | 9 | 33.3 |
| Services and sales worker | 9 | 33.3 |
| Student | 4 | 14.9 |
| Retired | 1 | 3.7 |
| National service personnel | 1 | 3.7 |
| Seamstress | 1 | 3.7 |
| House wife | 1 | 3.7 |
| **Number of children** | | |
| 0 | 13 | 48.2 |
| 1–2 | 8 | 29.6 |
| 3 and above | 6 | 22.2 |
| **Place of residence** | | |
| Urban | 26 | 96.3 |
| Rural | 1 | 3.7 |
| **History of non-communicable disease** | | |
| No non-communicable disease | 20 | 74.1 |
| Diagnosed with a non-communicable disease | 7 | 25.9 |
| **Total** | **27** | **100** |

belonged to the Christian religion. Furthermore, most participants resided in urban areas (96.3%) and had not been diagnosed with any non-communicable disease (74.1%). Finally, less than half of the participants (48.2%) had no children.

## Facilitators and barriers to adherence to COVID-19 preventive measures

Participants reported several facilitators and barriers to COVID-19 preventive measures. Table 2 shows the quotes of participants on facilitators and barriers to adherence to COVID-19 preventive measures.

**Adherence to COVID-19 preventive measures.**   All participants mentioned that they adhered strictly to the COVID-19 preventive measures, such as always wearing masks, washing hands with soap and water, and using alcohol-based hand sanitizer. However, they highlighted that persons dwelling in the communities no longer adhere to the COVID-19 preventive measures because they perceive that COVID-19 is no longer existing (Table 2 - Quote 1).

Participants also expressed that education about COVID-19 declined over time when COVID-19 cases reduced across the country (Quote 2). This situation may partly explain why some community members perceive COVID-19 no longer exists.

**COVID-19 preventive measures difficult to follow.**   According to the participants, the most challenging COVID-19 preventive measure to follow is wearing a face mask. This was followed by regular hand washing with soap under running water, social distancing, using a hand sanitizer, and touching one's face.

*Wearing a face mask*. Generally, most participants had difficulty always wearing face masks due to difficulty breathing in the face mask, tightness of the elastic ear loops of the face mask, heat inside the cover, and feeling uncomfortable and headache especially wearing it for a longer duration (Quotes 3 and 4). In addition, the participants expressed that their community members had similar experiences.

Despite the difficulty in wearing a face mask, some participants explained that wearing a face mask has become part of them, while others reported that purchasing face masks in town makes it easier to follow. Also, some participants narrated that using a face mask helps filter harmful/toxic gases from vehicles (Quote 5).

Furthermore, some participants explained that people are compelled to wear a face mask because it is a requirement to access certain services, including banking (Quote 6). Besides, failure to wear it could lead to one's arrest by the police service (Quote 7).

*Regular hand washing with soap under running water*. Participants reported that it is difficult to adhere to regular hand washing with soap under running water because of the following challenges: unavailability of water and tissue papers to use after washing hands, the dirtiness of the veronica buckets that contain the water for hand washing, forgetting to wash hands, and difficulty understanding why people must regularly wash their hands (Quotes 8 and 9).

However, some participants reported that regular hand washing with soap is easy to adhere to. The participants reported that they adhere to regular hand washing with soap under running water because hand washing is part and parcel of their lives (Quote 10). They also stated that regular hand washing with water under running water promotes personal hygiene and helps protect people from diseases (such as cholera) and microorganisms (including bacteria, fungi, and viruses) (Quote 11).

*Social distancing*. Social distancing also emerged as one of the COVD-19 preventive measures difficult to follow. Participants narrated that it is difficult to practice social distancing in markets, densely populated communities, lecture halls, and public commercial vehicles (Quotes 12 and 13).

**Table 2. Quotes on facilitators and barriers to adherence to COVID-19 preventive measures.**

| Quote ID | Sample quotes |
|---|---|
| Quote 1 | "My community members used to religiously adhere to the protocols when the disease [COVID-19] emerged. But they no longer adhere to the protocols since they think the disease has been eradicated." (R22) |
| Quote 2 | "Publicity on COVID-19 and education had decreased since it became less severe as compared to when it emerged in the country." (R29) |
| Quote 3 | "The face mask is difficult to use because it is sometimes uneasy wearing, especially when you wear it for a very long time. I think it gives a headache and makes one feel uncomfortable. So, it causes us to take it off once in a while." (R16) |
| Quote 4 | "The wearing of the face mask is difficult to follow because of the weather and the heat that generates inside the mask. I also struggle to breathe when I have the face mask on." (R28) |
| Quote 5 | "The face mask also filters some of the harmful or toxic gases coming from vehicles. So, even when COVID-19 had not emerged, I sometimes put on a face mask to prevent me from inhaling the harmful gases." (R25) |
| Quote 6 | "The wearing of face mask is easy because nowadays when you go to every place, like banks and stores, you will see the signage "no face mask, no entry." So, that compel people to wear it." (R13) |
| Quote 7 | "People find it easy to wear a face mask because the government brought a regulation that if you are not wearing the face mask, you would be arrested or the police would question you." (R6) |
| Quote 8 | "Regularly washing hands is difficult to follow because it is not every place that you will find a veronica bucket with soap and water to wash your hands. Sometimes, people do not remember to wash their hands all the time." (R6) |
| Quote 9 | "People do not understand why they should be washing their hands all the time. You have finished bathing, and you get out of the house to the bank, and you have to wash your hands. You move from the bank to the office or market, and you have to be washing your hands." (R26) |
| Quote 10 | "The washing of the hands is the easiest for me because that is already part of me, so I don't feel any pressure doing it." (R23) |
| Quote 11 | "The washing of hands with the soap under running water should be the easiest because it is something that even before COVID-19, we were doing. The washing of hands doesn't only prevent COVID-19, it also prevents so many diseases like cholera." (R27) |
| Quote 12 | "Social distancing is difficult to adhere to because I pick Trotro [public commercial vehicle] to work every day, and sometimes the bus conductor overloads the vehicle, so there is no way you can practice social distancing." (R4) |
| Quote 13 | "The only COVID-19 preventive measure that I am not observing is social distance, and it is because when I go for lectures, we are all together, and the tables and chairs are arranged in a way that we cannot practice social distancing." (R27) |
| Quote 14 | "When you don't practice certain things, doing it becomes difficult. I sometimes even forget that I have hand sanitizer in my bag." (R14) |
| Quote 15 | "I know some people have sanitizers in their bags that they use frequently. We, the market women, mostly forget to put ours in our bags. We get home before we use sanitizers." (R8) |
| Quote 16 | "Hand sanitizer is the easiest since most people have it in their bags since it is portable, so they tend to use it often, especially when they touch something." (R15) |
| Quote 17 | "I think using hand sanitizer is easy to follow because with the sanitizer; you can always use it when you touch something." (R16) |
| Quote 18 | "I think not touching faces will be difficult to follow because almost all the time it's either your hand is in your hair, your face, or you will be sweating, and you use your hands to clean the sweat from your hands." (R4) |
| Quote 19 | "It is challenging to avoid touching your face. Sometimes, a person sees pimples/acne on her face, and she may want to burst them. A person may see something on her face, and she may want to take it away from her face. It is not easy to stay away from touching one's face." (R6) |

*Using a hand sanitizer*. Some participants indicated that using a hand sanitizer is challenging because it is not part and parcel of people. Therefore, they sometimes leave their hand sanitizer at home, and even when they carry it along in their bag/purse, they sometimes forget to use them (Quotes 14 and 15).

However, some participants explained that using a hand sanitizer is easy because it is portable and thus easy to carry along (Quote 16). Also, some participants explained that hand sanitizer could easily be used or rubbed, especially in places where water is unavailable (Quote 17).

*Touching of one's face.* Touching one's face was also considered as one of the challenging COVID-19 preventive measures to follow. Participants narrated that it is challenging to avoid touching one's face since it is sometimes spontaneous (Quotes 18 and 19).

**Motivation to adhere to all COVID-19 preventive measures.** *Fear of getting infected.* Most participants adhered to the COVID-19 preventive measures to avoid being infected with the disease and associated morbidity (Table 3 - Quote 1). Some participants also revealed that they had witnessed the negative experiences of people infected with COVID-19 and thus were inspired to adhere to the preventive measures to avoid the consequences of contracting the disease (Quote 2).

In addition, the participants highlighted that they adhered to the COVID-19 preventive measures because they didn't want to be the people who would infect their family members with COVID-19 (Quote 3).

*Protection of one's health.* Protection of one's health also emerged as a motivation for adhering to COVID-19 preventive measures. Some participants emphasised that the desire to protect their health encouraged them to adhere to the COVID-19 preventive measures since their health is their priority (Quote 4).

## Discussion

Compliance with COVID-19 preventive measures is essential in reducing the spread of COVID-19. Using an in-depth interview, this study examined the barriers and facilitators of COVID-19 preventive measures among women in Ghana.

This study found that wearing a face mask emerged as the easiest COVID-19 preventive measure to follow because of its availability in stores, the benefit of filtering harmful/toxic gases and some institutions insist on wearing it before allowing entry (no mask, no entry). However, consistent with studies in Egypt, China, Indonesia and Iran [28, 54–56], key barriers include difficulty in breathing, feeling uncomfortable and headache, prevented some study participants and community members from wearing a face mask. Therefore, to encourage women to continue wearing a face mask, health practitioners need to educate them to realise that the benefits of wearing a face mask outweigh its costs. On March 27 2022, the wearing face masks was made voluntary in Ghana [57], and this directive could have implications for the spread of COVID-19.

Also, our results showed that regular hand washing with soap was one of the easiest COVID-19 preventive measures practiced by the participants and community members. According to the participants, individuals are motivated to regularly wash their hands because

**Table 3. Quotes on motivation to adhere to all COVID-19 preventive measures.**

| Quote ID | Sample quotes |
|---|---|
| Quote 1 | "My motivation is the fear of getting infected with COVID-19 since I might die if anything goes wrong." (R1) |
| Quote 2 | "A relative was infected with COVID-19, and he suffered a lot. I fear I may suffer like him when I am infected, so that motivated to adhere to the protocols." (R17) |
| Quote 3 | "My motivation is that I realised COVID-19 is real and it kills, so to protect myself from getting the virus and infecting members of my family, I follow the preventive measures." (R24) |
| Quote 4 | "What is motivating me is that my health comes first. Even if you have money and you are not healthy, you cannot do anything, so as for me, my health comes first." (R5) |

it is not a new culture. Before the emergence of COVID-19, hand washing with soap was practiced because of associated health benefits such as promoting personal hygiene and preventing diseases such as cholera. This emphasises the explanation of the socio-ecological model that cultural values promote health behaviours. As found in this study, protecting one's health is a key motivation to adhere to all the COVID-19 protocols. This finding is in line with other studies demonstrating that protecting one's health motivates individuals to adhere to COVID-19 preventive measures [42, 58]. Although regular hand washing with soap was considered easy, barriers, including unavailability of water, tissue papers, uncleanliness of veronica buckets, and forgetfulness, prevented some participants from practicing it. The unavailability of tissue paper is similar to a study by Fielmua et al.'s [26] in Ghana at shopping centres in Wa, which found that 88% (n = 50) of shops provided customers with no tissue papers to use after washing their hands. The authors explained that the unavailability of tissue paper was due to the high cost of tissue.

Also, the unavailability of water in veronica buckets, as found in our study, is similar to a study in Ethiopia which reported that lack of water was a barrier to the adherence to washing hands with soap under running water [33]. The unavailability of water in veronica buckets could be due to acute water shortage, which is a common phenomenon in Ghana and other sub-Saharan African countries [59–62]. Access to water in urban Ghana is zero for some residents and erratic for many others. This finding highlights the influence of structural level factors on health promoting behaviours.

The finding further indicated that practicing social distancing and using hand sanitizer is a new culture hence it is a challenge to adapt. The participants explained that because most people are not accustomed to the use of hand sanitizer, they mostly forget to use it even if they carry it along in their bags. This aligns with the community level of influence of the socio-ecological model impacting on health promoting behaviours. Studies in sub-Saharan Africa have reported poor compliance with social distancing protocol partly because of people's strong social ties and interaction. That is, people fear that practicing social distancing could negatively affect their cultural and social relationships [33, 63, 64].

Also, the participants stated that people do not comply with the regulation on social distancing due to issues, including the nature of the nation's transportation system, educational facilities and overcrowding areas and communities. In Ghana, the basic mode of transport for intracity travel among most commuters are public transport services popularly known as 'trotro'. These 'trotros' are popular compared to taxis cabs mainly because of their cheap fares. However, they are often congested and thus, increases the risk of spread of COVID-19 among passengers on board [43, 49]. During the initial stages of the COVID-19 pandemic in Ghana, precisely on 18th March 2020, the Ministry of Transport issued several directives for intercity travel to help prevent the spread of COVID-19. These included a reduction in the number of occupants per vehicle, a need for physical distancing in vehicles, an insistence on the washing of hands by passengers at bus stations before boarding vehicles, and the placement of locally devised wash sinks called veronica buckets at all bus stations [43]. However, currently all these measures are no longer enforced and this has implication on the spread of COVID-19.

## Limitations

The results presented in this study provide context of adherence to COVID-19 preventive measures among women. That is, this study provides evidence of the facilitators and barriers to the COVID-19 preventive measures among Ghanaian women living in urban areas.

However, the results of this study may not be generalisable to the entire population in two ways. First, results may not be representative because convenient sampling was used to select

respondents. Moreover, only views of women from two regions were included in the study–limiting the generalisability of results. Second, the study population has a potential selection bias since only non-pregnant women were eligible to participate in the study.

## Conclusions

This study highlights the drivers of compliance and hesitancy to COVID-19 preventive measures in Ghana. The main themes span the levels of health behaviour of the socio-ecological model of health. Interpersonal factors inhibited social distancing protocols though individuals would wish to adhere. Institutions that made adhering to COVID-19 measures a precondition for accessing service enhanced adherence. Societal and community norms, practices, and misconceptions about COVID-19 provide room for COVID-19 preventive measures non-compliance. These structural barriers, including inadequate access to water and crowded public transport services, contributed to individual level non-compliance, such as the handwashing guideline. The relative ease of accessing hand sanitizers and face masks enhanced their use among some study participants. Finally, government policies that mandated the population to wear face masks at all public places improved adherence. These multi-layered factors contribute to the overall behaviour towards COVID-19 preventive protocols. We, therefore, conclude that the fight against COVID-19 is not over; hence public education and the provision of facilities that would enhance compliance with preventive measures should continue to be prioritised.

## Supporting information

**S1 File. Interview guide administered to women.**
(DOCX)

## Acknowledgments

The authors would like to thank all women who took time out of their busy schedules to participate in the study. We are also grateful to all persons involved in the data collection for this study.

## Author Contributions

**Conceptualization:** Frank Kyei-Arthur, Martin Wiredu Agyekum, Grace Frempong Afrifa-Anane, Reuben Tete Larbi, Peter Kisaakye.

**Data curation:** Frank Kyei-Arthur, Martin Wiredu Agyekum, Grace Frempong Afrifa-Anane, Reuben Tete Larbi, Peter Kisaakye.

**Formal analysis:** Frank Kyei-Arthur.

**Investigation:** Frank Kyei-Arthur, Martin Wiredu Agyekum, Grace Frempong Afrifa-Anane, Reuben Tete Larbi.

**Methodology:** Frank Kyei-Arthur, Martin Wiredu Agyekum, Grace Frempong Afrifa-Anane, Reuben Tete Larbi.

**Project administration:** Frank Kyei-Arthur, Martin Wiredu Agyekum, Grace Frempong Afrifa-Anane, Reuben Tete Larbi.

**Resources:** Frank Kyei-Arthur, Martin Wiredu Agyekum, Grace Frempong Afrifa-Anane, Reuben Tete Larbi.

**Supervision:** Frank Kyei-Arthur, Martin Wiredu Agyekum, Grace Frempong Afrifa-Anane, Reuben Tete Larbi.

**Validation:** Frank Kyei-Arthur, Martin Wiredu Agyekum, Grace Frempong Afrifa-Anane, Reuben Tete Larbi, Peter Kisaakye.

**Writing – original draft:** Frank Kyei-Arthur, Martin Wiredu Agyekum, Grace Frempong Afrifa-Anane, Reuben Tete Larbi, Peter Kisaakye.

**Writing – review & editing:** Frank Kyei-Arthur, Martin Wiredu Agyekum, Grace Frempong Afrifa-Anane, Reuben Tete Larbi, Peter Kisaakye.

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
