## [Decision Letter · Decision Letter 0]

16 Jan 2023

PONE-D-22-15561Perceptions about COVID-19 preventive measures among Ghanaian womenPLOS ONE

Dear Dr. Kyei-Arthur,

Thank you for submitting your manuscript to PLOS ONE. After careful consideration, we feel that it has merit but does not fully meet PLOS ONE’s publication criteria as it currently stands. Therefore, we invite you to submit a revised version of the manuscript that addresses the points raised during the review process.

In addition, the methods section of the study requires review. This is a qualitative study however the design and method used for data analysis is not stated explicitly and not justified. 'Cross-sectional qualitative study' is not an appropriate name; method of sample selection needs to be clearly stated with  justification. I recommend the authors use the COREQ check list to guide the reporting of this study. 

Please submit your revised manuscript within 15days. If you will need more time than this to complete your revisions, please reply to this message or contact the journal office at plosone@plos.org. Please include the following items when submitting your revised manuscript:A rebuttal letter that responds to each point raised by the academic editor and reviewer(s). You should upload this letter as a separate file labeled 'Response to Reviewers'.A marked-up copy of your manuscript that highlights changes made to the original version. You should upload this as a separate file labeled 'Revised Manuscript with Track Changes'.An unmarked version of your revised paper without tracked changes. You should upload this as a separate file labeled 'Manuscript'.

We look forward to receiving your revised manuscript.

Kind regards,

Ogochukwu Chinedum Okoye

Academic Editor

PLOS ONE

Journal Requirements:

Additional Editor Comments (if provided):

Reviewers' comments:

Reviewer's Responses to Questions

**Comments to the Author**

1. Is the manuscript technically sound, and do the data support the conclusions?

Reviewer #1: Yes

Reviewer #2: Yes

2. Has the statistical analysis been performed appropriately and rigorously? 

Reviewer #1: Yes

Reviewer #2: Yes

3. Have the authors made all data underlying the findings in their manuscript fully available?

Reviewer #1: Yes

Reviewer #2: Yes

4. Is the manuscript presented in an intelligible fashion and written in standard English?

Reviewer #1: Yes

Reviewer #2: Yes

5. Review Comments to the Author

Reviewer #1: The study is a valuable addition to scientific literature as it used qualitative design. Many studies are available on the subject but only a few utilize interview based approach and present data in that form which is a better reflection of actual barriers that are perceived by the population.

Reviewer #2: Question 1. The manuscript is technically sound

Question 2. Qualitative data analysis was done using Nvivo 10. software as reported by the authors. The data analysis process was well outlined. However authors should note the following:

Page 7, line 153: report the standard deviation and not just the mean age of the respondents under the socio- demographic characteristics. Line 155: is there a reference to the categorisation scheme for occupations highlighted in this study or was it arbitrary.

Page 11, line 177: is the term "dominance" synonymous with frequency of response or agreement

Pages 11 and 12: the reporting of COVID 19 preventive measures as easy to follow versus COVID 19 preventive measures difficult to follow is confusing especially when noting that "wearing a facemask" features as number one in both categories. One would have better understood a spectrum of preventive measures from the easiest to the most difficult. The authors should clarify on this. Another approach recommended could be to take the individual COVID 19 preventive measures and express the relative ease or difficulty in application of each as reported by respondents.

Page 14, Table 3. Quote 2 "As a frontline worker . . . (R11)"

(Should a frontline ?Health worker, R11, be included in your sample for this study, knowing that her knowledge and job requirement would bias the level of knowledge and lived experience that would be elicited from her compared to other respondents? Can the authors justify her inclusion)

Question 3. Yes, as disclosed by the authors in the "Data Availability" section.

Question 4. Yes, the manuscript was reported in Standard English. However the manuscript will benefit from editing such as:

Abstract

Page 1, line 39: insert "should" e.g ' . . . measures should continue to be prioritised".

Introduction

Page 2, line 60, 61: rephrase ". . . SARS-CoV-2 virus can escape immunity causing infection . . ."

Methods

Page 5, line 109: The sampling procedure is not clearly explained by the authors. What is "convenient" and "snowballing" sampling procedure in the context of the current study? What was the study population and what was so peculiar in recruiting the subjects of the study necessitating the use of snowball sampling?

6. PLOS authors have the option to publish the peer review history of their article (what does this mean?). If published, this will include your full peer review and any attached files.

Reviewer #1: No

Reviewer #2: No

---

## [Author Response · Author response to Decision Letter 0]

20 Jan 2023

Academic editor 

1. The methods section of the study requires review. This is a qualitative study however the design and method used for data analysis is not stated explicitly and not justified. 'Cross-sectional qualitative study' is not an appropriate name; method of sample selection needs to be clearly stated with justification. I recommend the authors use the COREQ check list to guide the reporting of this study. 

Response: We have revised the study design and sampling procedure to make study design and sampling procedure understandable. We have also defined convenient sampling procedure in the revised manuscript. See page 5 and 6, lines 108-122.

The method of data analysis was clearly stated in the manuscript. See page 7, lines 153-154. 

“The data for the study were analysed thematically using NVivo 10 software following steps recommended by Braun and Clarke [52, 53].” 

We have justified why we followed Braun and Clarke recommended steps for analysing qualitative data in the revised manuscript. See page 8, lines 162-164.

“Braun and Clarke recommended steps for analysing qualitative data were followed because it helps researchers to analyse qualitative data systematically and in a rigorous manner.”

We have revised the entire methods section and used the COREQ checklist to revise the reporting of the study.

Reviewer 1

1. Reviewer #1: The study is a valuable addition to scientific literature as it used qualitative design. Many studies are available on the subject but only a few utilize interview based approach and present data in that form which is a better reflection of actual barriers that are perceived by the population.

Response: We thank the reviewer for the complimentary words. 

Reviewer 2

1. Page 7, line 153: report the standard deviation and not just the mean age of the respondents under the socio- demographic characteristics.

Response: We have reported both the mean age and standard deviation of participants in the revised manuscript. See Table 1 on page 9. 

2. Line 155: is there a reference to the categorisation scheme for occupations highlighted in this study or was it arbitrary.

Response: The categorisation scheme for occupations was arbitrary based on occupations reported by participants.

3. Page 11, line 177: is the term "dominance" synonymous with frequency of response or agreement.

Response: In this study, the term "dominance" is synonymous with frequency of response or agreement.

4. Pages 11 and 12: the reporting of COVID 19 preventive measures as easy to follow versus COVID 19 preventive measures difficult to follow is confusing especially when noting that "wearing a facemask" features as number one in both categories. One would have better understood a spectrum of preventive measures from the easiest to the most difficult. The authors should clarify on this. Another approach recommended could be to take the individual COVID 19 preventive measures and express the relative ease or difficulty in application of each as reported by respondents.

Response: We thank the reviewer for the suggestions. We have revised the results on the COVID-19 preventive measures by reporting the preventive measures by their relative difficulty. While reporting the relative difficulty of each preventive measure, we reported on their easiness where applicable. We have also revised the quotes in Table 2 since the quote numbers changed. See pages 12-14 lines 200-238, Table 2. 

5. Page 14, Table 3. Quote 2 "As a frontline worker . . . (R11)"

(Should a frontline ?Health worker, R11, be included in your sample for this study, knowing that her knowledge and job requirement would bias the level of knowledge and lived experience that would be elicited from her compared to other respondents? Can the authors justify her inclusion)

Response: The knowledge and job requirement of health workers will bias their level of knowledge and lived experiences. Consequently, three participants were health workers, so we have excluded them from the study. We have changed the sample size in the revised manuscript from 30 to 27. Consequently, we have revised the characteristics of participants, and replaced the quotes of participants who were health workers with quotes of participants who are non-health workers. See page 8, lines 171-177 and Table 1, 2 and 3. 

6. Question 4. Yes, the manuscript was reported in Standard English. However the manuscript will benefit from editing such as:

Abstract

Page 1, line 39: insert "should" e.g ' . . . measures should continue to be prioritised".

Response: We have revised the phrase “measures continue to be prioritised” to “measures should continue to be prioritised”. See page 2, line 40. 

7. Introduction

Page 2, line 60, 61: rephrase ". . . SARS-CoV-2 virus can escape immunity causing infection . . ."

Response: We have revised the sentence to “Also, studies have established that SARS-CoV-2, the COVID-19-causing virus, can escape immunity, which can cause infection among people or reinfection among people who have already been vaccinated against COVID-19 or previously been infected with COVID-19 [6, 12-14]”. See 3, lines 60-63. 

In addition, we have edited the entire manuscript to address any grammatical error.

8. Methods: Page 5, line 109: The sampling procedure is not clearly explained by the authors. What is "convenient" and "snowballing" sampling procedure in the context of the current study? 

Response: We have revised the study design and sampling procedure to make sampling procedure understandable. We have also defined convenient sampling procedure in the revised manuscript. See page 5 - 6, lines 108-122.

9. What was the study population and what was so peculiar in recruiting the subjects of the study necessitating the use of snowball sampling?

Response: The study population for the study are non-pregnant women aged 18 years or older in the Greater Accra and Ashanti Regions. We have indicated it in the revised manuscript on pages 5-6 lines 112-113. 

Also, this study used only convenient sampling rather than convenient and snowball sampling procedures. Therefore, we have deleted snowball sampling from the revised manuscript.

---

## [Decision Letter · Decision Letter 1]

29 Mar 2023

Perceptions about COVID-19 preventive measures among Ghanaian women

PONE-D-22-15561R1

Dear Dr. Kyei-Arthur,

We’re pleased to inform you that your manuscript has been judged scientifically suitable for publication and will be formally accepted for publication once it meets all outstanding technical requirements.

Kind regards,

Dario Ummarino, PhD

Senior Editor

PLOS ONE

Additional Editor Comments (optional):

Reviewers' comments:

Reviewer's Responses to Questions

**Comments to the Author**

1. If the authors have adequately addressed your comments raised in a previous round of review and you feel that this manuscript is now acceptable for publication, you may indicate that here to bypass the “Comments to the Author” section, enter your conflict of interest statement in the “Confidential to Editor” section, and submit your "Accept" recommendation.

Reviewer #1: All comments have been addressed

Reviewer #2: All comments have been addressed

2. Is the manuscript technically sound, and do the data support the conclusions?

Reviewer #1: Yes

Reviewer #2: (No Response)

3. Has the statistical analysis been performed appropriately and rigorously? 

Reviewer #1: Yes

Reviewer #2: (No Response)

4. Have the authors made all data underlying the findings in their manuscript fully available?

Reviewer #1: Yes

Reviewer #2: (No Response)

5. Is the manuscript presented in an intelligible fashion and written in standard English?

Reviewer #1: Yes

Reviewer #2: (No Response)

6. Review Comments to the Author

Reviewer #1: The manuscript can be accepted in it's present form . The authors have satisfactorily answered all the points raised by reviewer 2 in my humble opinion.

Reviewer #2: (No Response)

7. PLOS authors have the option to publish the peer review history of their article (what does this mean?). If published, this will include your full peer review and any attached files.

Reviewer #1: No

Reviewer #2: **Yes: **Dr. Nyemike Simeon Awunor

---

## [Editor Report · Acceptance letter]

3 Apr 2023

PONE-D-22-15561R1 

Perceptions about COVID-19 preventive measures among Ghanaian women 

Dear Dr. Kyei-Arthur:

I'm pleased to inform you that your manuscript has been deemed suitable for publication in PLOS ONE. Congratulations! Your manuscript is now with our production department. 

Kind regards, 

on behalf of

Dr Dario Ummarino, PhD 

Staff Editor

PLOS ONE